# Structural Insight into Integrin Recognition and Anticancer Activity of Echistatin

**DOI:** 10.3390/toxins12110709

**Published:** 2020-11-09

**Authors:** Yi-Chun Chen, Yao-Tsung Chang, Chiu-Yueh Chen, Jia-Hau Shiu, Chun-Ho Cheng, Chun-Hao Huang, Ju-Fei Chen, Woei-Jer Chuang

**Affiliations:** Department of Biochemistry and Molecular Biology, National Cheng Kung University College of Medicine, Tainan 70101, Taiwan; analskan@gmail.com (Y.-C.C.); ytchang.ncku@gmail.com (Y.-T.C.); cyccyc168@gmail.com (C.-Y.C.); jiahau.shiu@gmail.com (J.-H.S.); chunho.cheng@gmail.com (C.-H.C.); qbi21@yahoo.com.tw (C.-H.H.); dfeisfayfay@gmail.com (J.-F.C.)

**Keywords:** echistatin, disintegrins, crystal structure, cancer cell migration, HUVEC proliferation, anticancer drug design

## Abstract

Echistatin (Ech) is a short disintegrin with a long ^42^NPHKGPAT C-terminal tail. We determined the 3-D structure of Ech by X-ray crystallography. Superimposition of the structures of chains A and B showed conformational differences in their RGD loops and C-termini. The chain A structure is consistent with our NMR analysis that the GPAT residues of the C-terminus cannot be observed due to high flexibility. The hydrogen bond patterns of the RGD loop and between the RGD loop and C-terminus in Ech were the same as those of the corresponding residues in medium disintegrins. The mutant with C-terminal HKGPAT truncation caused 6.4-, 7.0-, 11.7-, and 18.6-fold decreases in inhibiting integrins αvβ3, αIIbβ3, αvβ5, and α5β1. Mutagenesis of the C-terminus showed that the H44A mutant caused 2.5- and 4.4-fold increases in inhibiting αIIbβ3 and α5β1, and the K45A mutant caused a 2.6-fold decrease in inhibiting αIIbβ3. We found that Ech inhibited VEGF-induced HUVEC proliferation with an IC_50_ value of 103.2 nM and inhibited the migration of A375, U373MG, and Panc-1 tumor cells with IC_50_ values of 1.5, 5.7, and 154.5 nM. These findings suggest that Ech is a potential anticancer agent, and its C-terminal region can be optimized to improve its anticancer activity.

## 1. Introduction

Echistatin (Ech) is a short disintegrin isolated from *Echis carinatus* that contains 49 residues with 4 disulfide bonds. Ech is the most extensively characterized disintegrin and has a conserved RGD motif and a long C-terminal NPHKGPAT tail [1,2,3,4,5,6]. Several studies support a functional role for the C-terminal residues of Ech in modulating its binding affinity and selectivity towards integrin αIIbβ3 [6,7,8]. For example, a synthetic C-terminal peptide of Ech inhibited the binding of integrin αIIbβ3 to Ech, activated the binding of integrin to immobilized ligands, induced the expression of activation-dependent conformational epitopes of integrin, and increased the binding of fibrinogen to peptide-treated platelets [6,7,9,10]. An RGD peptide fused with the C-terminal KGPAT residues of Ech selectively binds αvβ3 integrin but not αvβ5 integrin [11]. Replacement of contortrostatin’s C-terminal FHA with Ech’s C-terminal HKGPAT tail results in a higher binding affinity to integrin α5β1 [12]. In our previous study, we also demonstrated that the length of the C-terminus and the K45 residue of Ech are important for its interactions with integrin αIIbβ3 [13]. These results show that the C-terminal tail of Ech may act in synergy with the RGD loop to interact with integrins. However, little is known about the role of the C-terminus in recognizing RGD-dependent integrins.

Many 3-D structures of disintegrins have been determined by X-ray crystallography and NMR spectroscopy [6,14,15,16,17]. Comparison of the 3-D structures of disintegrins reveals a very similar fold with an elongated shape [14,15,16,17,18]. Their secondary structures are composed of a series of short antiparallel β-sheets. The RGD motif is located at the apex of the integrin-binding loop, which is structurally close to the C-terminus. The 3-D structure of Ech has also been determined by NMR spectroscopy [3,6,13,19]. Similar to the reported structures of other disintegrins, Ech has a well-defined core composed of a series of nonclassical turns crosslinked by disulfide bonds (C2-C11, C7-C32, C8-C37, and C20-C39). The RGD motif is located in a protruding loop that is formed by two antiparallel β-strands (residues 16–20 and 29–31). A dynamics study showed that the RGD loop and the C-terminal region exhibited concerted motion [6]. This is consistent with functional studies showing that the C-terminus of Ech plays an important role in integrin recognition [6,7,8,9,10,11,12,13]. However, the C-terminal region of Ech is flexible based on the reported NMR structure [3,6,19]. It is essential to determine the structural conformation of Ech’s C-terminus in a high-resolution crystal structure to elucidate its structure–function relationships.

Integrins have been found to be associated with tumor angiogenesis, tumor cell migration, proliferation, and metastasis [20,21,22,23]. Disintegrins are potent integrin inhibitors [24,25,26,27,28,29,30,31,32,33]. Many in vitro and in vivo studies have demonstrated that RGD-containing disintegrins have antiangiogenic and antimetastatic activities [24,25,26,27,28,29,30,31,32,33]. They are effective in inhibiting angiogenesis, tumor cell adhesion, and tumor cell migration with micromolar to high nanomolar potency [24,25]. Ech also significantly inhibited angiogenesis, lung metastasis of B16F10 melanoma, metastasis of MV3 melanoma, and the hyaluronan-mediated migration of glioblastoma cells in fibrin gels [26]. To evaluate the inhibitory activities of recombinant Ech and its mutants, we determined the IC_50_ values of VEGF-induced HUVEC proliferation and tumor cell migration, as well as their activities in inhibiting integrins αvβ3, αvβ5, and α5β1.

To analyze how the C-terminus of Ech interacts with the RGD loop to recognize integrins, we determined its 3-D structure using X-ray crystallography. Site-directed mutagenesis and deletion studies on Ech were used to identify the essential residues involved in the recognition of integrins, as well as its antiangiogenic and antimetastatic activities. We found that the major differences in disintegrins were the amino acid composition, length, and conformation of the C-terminus. These structural and functional studies of Ech serve as the basis for insight into their structure–function relationships and for the design of anticancer drugs.

## 2. Results

### 2.1. The Overall Structure of Ech

The diffraction data of Ech were obtained to a resolution of 1.80 Å from a single frozen crystal, which contained two molecules per asymmetric unit. Figure 1A–D show a representative electron-density fit including the RGD loop from residues K21 to D30 and the C-terminus from residues P40 to T49. Ramachandran plot analysis of the final structure, obtained from the program MolProbity [34], showed 95.7% of the residues in the favored region, 4.3% in the additionally allowed region, and no residues in the outlier region (Table 1). The data collection and structure refinement statistics are summarized in Table 1. The overall structure of chains A and B contained four disulfide bonds (C2-C11, C7-C32, C8-C37, and C20-C39) and one pair of antiparallel β-strands (residues 16–20 and 29–31) (Figure 2A,B).

Superimposition of the backbones of chains A and B with an RMSD value of 0.226 Å demonstrated that their overall structures were very similar, except for the RGD loop and C-terminal loop (Figure 2C–E). Superimposition of the RGD loop (residues K21-D30) and C-terminus (residues P40-K45) resulted in RMSD values of 0.162 Å and 0.581 Å, respectively (Figure 2D,E). Structural analysis showed that the RGD loop of chain B was squeezed by residues K21, R41, and K45 in contact with sulfate ions and residue S4 of chain A through crystal packing (Appendix A). Therefore, the R and D side chains of the chain B structure interacted with each other and formed hydrogen bonds (Figure 2D). In contrast to chain B, the R and D side chains in the RGD loop of chain A are far from each other (Figure 2D). Furthermore, the backbones of the C-terminal GPAT residues of chain A showed a weaker electron density map (Figure 1C). This was consistent with our unpublished NMR data and NMR nuclear Overhauser effect (NOE) analysis, which showed that the GPAT residues of the C-terminus exhibited low or negative NOE values (Appendix A). The average NOE value of Ech is 0.605. In addition, residues R24-D27 in the RGD loop, E1-E3 in the N-terminus, and K45-T49 in the C-terminus had NOE values less than the average NOE value of Ech, suggesting that these regions of Ech were flexible. In contrast, the conformation of the C-terminal tail of Ech chain B was well defined (Figure 1D), possibly because the C-terminal residue K45 can interact with the O atoms of SO_4_^2-^ (Appendix A). Moreover, the C-terminal residue T49 can interact with water molecules (Appendix A). 

Although the tertiary fold of Ech was similar to the reported NMR structure (2Ech), superimposition of the backbones of Ech chain A and the NMR structure (2Ech) resulted in RMSD values of 1.688 Å (Figure 2F,G) [13]. Nevertheless, the analyses of their structures showed that the RGD motif is located at the apex of the integrin-binding loop, which is structurally close to the C-terminus. These findings were consistent with the reported 3-D structures of medium disintegrins [14,15,16,17,18]. The resulting high-resolution crystal structure can provide an improved molecular basis to design anticancer agents using Ech as a scaffold.

### 2.2. Structural Comparison of the RGD Loop and the Interactions between the RGD Loop and C-Termini

Hydrogen bond connectivity and atom distances of the RGD loop were compared among Ech chains A and B and trimestatin, a medium disintegrin with an ARGDFP motif, to identify their structural differences in detail (Figure 3). We found that the Cα to Cα distance between residues R and D in chain A, chain B, and trimestatin of the RGD motif was 6.5, 5.8, and 6.7 Å, and the Cζ to Cγ distance between residues R and D in chain A, chain B, and trimestatin was 12.4, 4.8, and 11.1 Å (Figure 3A–C, Table 2). These results showed that Ech chain A and trimestatin exhibited similarity in the conformation of the RGD motif and in the sidechain orientation of the R and D residues, which were pointed away from each other. Moreover, comparison of the Cα(Ri)-Cα(Di+2), Cβ(Ri)-Cβ(Di+2), Cζ(Ri)-Cγ(Di+2), and Cα(Ri)-Cα(Xi+3) distances (Å) of Ech chain A and Ech’s NMR structure (2Ech) revealed that the RGD motif in our X-ray structure (Ech chain A) was similar to that in the solution NMR structure (2Ech) (Table 2). Interestingly, we found a conserved and extensively connected hydrogen bond network in the RGD loop. The residues located on both sides of the RGD motif in Ech formed hydrogen bonds, including K21 with D30, A23 with D29, and R24 with D27 (Figure 3D,E, Appendix A). The corresponding residues of trimestatin also formed hydrogen bonds, including R46 with D55 and A48 with D54 (Figure 3F, Appendix A). These results showed that short and medium disintegrins have similar RGD motif and loop conformations, suggesting that these conserved interactions may be important to integrin recognition because they maintain the conformation of the RGD loop.

Similar to the conformation of the RGD loop, the interactions between the RGD loop and C-terminus were conserved in short and medium disintegrins (Figure 4A–C, Appendix A). The structural analysis of Ech chains A and B showed that the side chain of the D30 residue in the RGD loop and the amide proton of the N42 residue in the C-terminus formed hydrogen bonds (Figure 4A,B). The corresponding residues of trimestatin also formed hydrogen bonds between the side chain of residue D55 in the RGD loop and the amide proton of residue W67 (Figure 4C). In contrast to Ech chain A and trimestatin, Ech chain B formed additional hydrogen bonds, including residue M28 with residue K45 and residue D30 with residue K45 (Figure 4B). These results implied that the longer C-terminal tail, which forms the loop-like structure, may play an important role in the interaction with integrins.

### 2.3. Structural Comparison of the C-Terminal Regions 

In addition to the similar conformation of the RGD loop and the interactions between the RGD loop and C-terminus, we also found that short and medium disintegrins shared a similar interaction in their C-termini (Figure 4D–F, Appendix A). The N42 residue can interact with the mainchain amide proton of the P43 residue in the Ech chain A (Figure 4D) and chain B (Figure 4E). The corresponding residues of trimestatin also formed hydrogen bonds between the sidechain amide proton of residue W67 and the carboxyl group of residue N68 (Figure 4F). Thus, we know that these conserved regions of the C-terminus are essential for its structure. Furthermore, in contrast to trimestatin, the extended C-terminal tail residue H44 can interact with N42 in Ech chain A (Figure 4D) and chain B (Figure 4E). These results showed that the conformation of the C-terminal regions of Ech may be important in integrin recognition.

### 2.4. Structural Differences between the Integrin Complexes of Ech Chain A and Chain B 

The docking of Ech chain A and chain B into integrin αvβ3 was used to identify the integrin-interacting residues. The analysis showed that the docking of Ech chain A and chain B into integrin αvβ3 resulted in different binding modes, especially in the C-terminus. (Figure 5A,B). The docking of Ech chain A into integrin αvβ3 showed that the side chain of the R22 residue of Ech chain A may interact with the main chain of N313 in the β3 subunit and that the main chain of the G25 residue of Ech chain A can also interact with the side chain of D218 in the αv subunit (Table 3 and Appendix A). In contrast, Ech chain B did not show these kinds of interactions but instead interaction between the side chain of M28 and the main chain of D126 in the β3 subunit (Table 4 and Appendix A). On the other hand, the main chain of C-terminal residue H44 of Ech chain A can interact with Y122 of the β3 subunit, and this contact was not found in Ech chain B. However, the side chain of C-terminal residue K45 of Ech chain B can interact with the side chain of M180 of the β3 subunit. These results showed that the C-terminal tails of Ech chains A and B were involved in Ech and β3 subunit interactions. These results suggested that Ech with a longer C-terminal tail may increase its recognition site and binding affinity to αvβ3 integrin.

### 2.5. The C-Terminal Residues of Ech are Involved in Mediating Integrin Recognition

The lengths of the C-terminus and the K45 residue of Ech were found to be important for interacting with integrin αIIbβ3 [13]. In this study, the inhibition of the binding of cell-expressed integrins αvβ3, αIIbβ3, αvβ5, and α5β1 to their ligands by Ech and its mutants was used to determine their activity and selectivity (Table 5 and Table 6). Compared to wild-type Ech, the mutant with the truncation of the C-terminal HKGPAT tail caused 6.4-, 7.0-, 11.7-, and 18.6-fold decreases in the inhibition of integrins αvβ3, αIIbβ3, αvβ5, and α5β1, respectively (Table 5). The effect of the C-terminal region on the change in the relative binding affinity to integrins was α5β1 > αvβ5 > αIIbβ3 > αvβ3. These results indicated that the C-terminal tail of Ech played an important role in recognizing not only integrin αIIbβ3 but also integrins αvβ3, αvβ5, and α5β1 [7,10,13].

Alanine scanning mutagenesis of the C-terminal tail of Ech showed that wild-type, P43A, H44A, and K45A mutants exhibited similar activity in inhibiting the adhesion of CHO cells that expressed integrin αvβ3 to immobilized fibrinogen with IC_50_ values of 20.7, 13.8, 16.4, and 14.7 nM (Table 6). In contrast to integrin αvβ3, some differences were found in their activities in inhibiting integrins αIIbβ3 and α5β1. They inhibited the adhesion of CHO cells that expressed integrin αIIbβ3 to immobilized fibrinogen with IC_50_ values of 51.5, 67.7, 20.7, and 132.5 nM and inhibited K562 cell adhesion to immobilized fibronectin with IC_50_ values of 132.6, 58.5. 30.0, and 84.9 nM. Interestingly, the H44A mutant caused 2.5- and 4.4-fold increases in the inhibition of αIIbβ3 and α5β1, and the K45A mutant caused a 2.6-fold decrease in the inhibition of integrin αIIbβ3. These results suggest that residues H44 and K45 may play different roles in integrin recognition. Indeed, in our docking structure of integrin αvβ3 with Ech chain A, we found that the main chain of C-terminal residue H44 in Ech chain A formed a hydrogen bond with the side chain of residue Y122 in the β3 subunit (Figure 5A, Table 3 and Appendix A). Moreover, in the docking structure of integrin αvβ3 with Ech chain B, we found that the sidechain of C-terminal residue K45 in Ech chain B formed a hydrogen bond with the side chain of residue M180 in the β3 subunit (Figure 5B, Table 4 and Appendix A). Our findings suggest that a longer C-terminal tail may improve the recognition site of Ech and increase its binding affinity to αvβ3 integrin, endowing Ech with inhibitory activity against αvβ3 integrin-related cancer. In the future, we can apply this property in Ech-based drug design for cancer therapy.

### 2.6. Inhibition of VEGF-Induced HUVEC Proliferation by Ech

Angiogenesis is a critical process for cancer cell growth and metastasis [35]. Endothelial cell proliferation is the key step in angiogenesis progression. Many studies have shown that disintegrins are potent angiogenic inhibitors [25,26,27,28,29,30,31,32]. The results of our analysis showed that Ech could inhibit HUVEC proliferation with an IC_50_ value of 103.2 nM (Figure 6, Table 7). In comparison with the reported disintegrin with IC_50_ values of 100–400 nM [17,30,31,32], Ech exhibited similar activity in inhibiting HUVEC proliferation (Table 7). These results suggest that Ech could be a good scaffold to design potent antiangiogenic agents.

### 2.7. C-Terminal Residue K45 of Ech Is Responsible for Its Anti-Metastatic Ability

Many studies have shown that disintegrins inhibit the adhesion and migration of tumor cells [24,25,26,29]. In our previous study, we also demonstrated that recombinant Ech effectively inhibited human A375 melanoma with an IC_50_ value of 1.5 nM (Table 8) [13]. To study the inhibitory effect of Ech on the migration of other tumor cells, we examined its antimigration activities in human glioblastoma U373MG and pancreatic Panc-1 cells. Similar to the results of melanoma cell inhibition by Ech, it inhibited the migration of U373MG and Panc-1 tumor cells with IC_50_ values of 5.7 and 154.5 nM, respectively (Figure 7 and Figure 8, Table 8). We found that the K45E mutant exhibited 8.9-, 4.8-, and 10.4-fold decreases in inhibiting the migration of A375, U373MG, and Panc-1 tumor cells (Figure 7 and Figure 8, Table 8), showing the importance of the K45 residue in inhibiting tumor cell migration. According to our X-ray structure, the conformation of Ech’s C-terminus was stabilized by the carboxyl group of residue M28 and the side-chain Nζ proton of residue K45 and between the side chain of residue D30 and the side-chain Nζ proton of residue K45 (Figure 4B). Furthermore, our docking structure of integrin αvβ3 with Ech chain B also showed that the side chain of C-terminal residue K45 in Ech chain B can interact with the side chain of residue M180 in the β3 subunit (Figure 5B, Table 4 and Appendix A). These findings suggested that the conformation of the C-terminus plays an important role in αvβ3 integrin recognition, resulting in the inhibition of tumor cell migration.

## 3. Discussion

RGD-containing disintegrins have potent antiangiogenic and antimetastatic activities [23,24,25,26,27,28,29,30,31,32]. To elucidate the structure–function relationships of Ech, we determined the crystal structure of Ech and its anti-integrin, antiangiogenic, and antimigration activities. We found that the conformation of the RGD loop and the interactions between the RGD loop and C-terminus were similar to those in the reported medium disintegrin trimestatin [15]. In contrast, significant differences between short and medium disintegrins were found in their C-terminal conformations. We also showed that the C-terminus of Ech plays important roles in integrin recognition and the inhibition of tumor cell migration. This structural and functional study of Ech can serve as a basis for insight into its structure–function relationships and for the design of anticancer drugs.

Ech is the most studied disintegrin, and Aggrastat (Tirofilban), an antiplatelet drug, was designed based on its RGD loop structure [33]. It is known that the RGD loop, linker, and C-terminal regions of disintegrins are involved in integrin recognition [18,25,36]. The incorporation of C-terminal structural information into current drugs may improve its potency and selectivity [37]. Previous studies also showed that the C-terminal residues of Ech possess a functional role in modulating the binding affinity to integrin αIIbβ3 [6,7,8,9,10,13]. The analysis of Ech’s crystal structure showed that the sidechain of the D30 residue in the RGD loop and the amide proton of the N42 residue in the C-terminus formed hydrogen bonds. This is consistent with the report that the C-terminal tail of Ech acts in synergy with the RGD loop to interact with integrins [6,7,10,13]. According to our X-ray crystal structure and NMR NOE analysis, the residues from G46 to T49 of the C-terminus are flexible and cannot be observed in the X-ray structure. Disordered regions of proteins frequently have important biological functions. Interestingly, the carboxyl group of residue M28 interacts with the side-chain Nζ proton of residue K45, and the side chain of residue D30 interacts with the side chain Nζ proton of residue K45, resulting in a unique turn structure for Ech. These results showed that the K45 residue stabilizes the conformation of Ech’s C-terminus and is important for its integrin recognition. The unique conformation of Ech’s C-terminus can be used to design integrin-specific drugs.

Alanine scanning mutagenesis of the C-terminal tail of Ech showed that the H44A mutant caused 2.5- and 4.4-fold increases in the inhibition of αIIbβ3 and α5β1, and the K45A mutant caused a 2.6-fold decrease in the inhibition of integrin αIIbβ3. These results suggested that residues H44 and K45 may play different roles in integrin recognition. Moreover, we found a correlation between our docking results and these functional studies. In Ech chain A, the main chain of C-terminal residue H44 formed a hydrogen bond with the side chain of residue Y122 in the β3 subunit (Figure 5A, Table 3 and Appendix A). On the other hand, in Ech chain B, the sidechain of C-terminal residue K45 interacts with the sidechain of residue M180 in the β3 subunit (Figure 5B, Table 4 and Appendix A). The different interaction modes between H44/K45 and αvβ3 integrin may provide information on how H44/K45 binds to other integrins.

Integrins are highly expressed in many cancers, including melanoma, pancreatic cancer, glioblastoma, breast cancer, prostate cancer, ovarian cancer, and cervical cancer, and are involved in cancer cell progression [20,21,22,23]. A number of studies have reported that Ech is a potent antagonist of integrins αvβ3, αIIbβ3, and α5β1 for cancer therapy and inhibits HUVEC adhesion to immobilized fibronectin and vitronectin, as well as cell proliferation, migration, invasion, and adhesion of metastatic human osteosarcoma [6,10,24,25,26,38]. T cells genetically engrafted with Ech-containing chimeric antigen receptor could also efficiently inhibit integrin-expressing solid tumors [39]. We showed that Ech could inhibit VEGF-induced HUVEC proliferation with an IC_50_ value of 103.2 nM and inhibit the migration of human melanoma A375, glioblastoma U373MG, and pancreatic Panc-1 tumor cells with IC_50_ values of 1.5, 5.7, and 154.5 nM. These findings indicate that Ech is a potent antiangiogenic and antimetastatic agent. The resulting structure–function relationships found in this study can provide a molecular basis for the use of Ech as a molecular scaffold to design anticancer drugs in the future.

## 4. Conclusions

In conclusion, this is the first report to determine a high-resolution crystal structure of a short disintegrin with a resolution of 1.80 Å. In comparison with the crystal structure of medium disintegrin, they have similar conformations for the RGD loop and the same hydrogen bond connectivity between the RGD loop and C-terminus. The major difference was found from the conformation of their C-termini. According to the results of the functional analysis, Ech may be a potential candidate for anticancer therapeutics with antiangiogenic and antimigration properties, and the C-terminal region of Ech can be optimized to improve its anticancer activity. The structure–function relationships of Ech can provide a molecular basis to design anticancer drugs using Ech as a molecular scaffold.

## 5. Materials and Methods 

### 5.1. Expression, Purification, and Characterization of Ech and Its Mutants 

The expression and purification of Ech and its mutants in *P. pastoris* were performed following previously described protocols [13,14,18,40,41,42]. In brief, Ech and its mutants were expressed in the *P. pastoris* X33 strain with a 6xHis-tag, and purified to apparent homogeneity by Ni^2+^-chelating chromatography and HPLC. Wild-type Ech without a 6xHis-tag was expressed for crystallization and purified to homogeneity by CaptoMMC chromatography and HPLC. The yields of Ech and its mutants produced in *P. pastoris* were 2–7 mg/L. Mass spectrometry analysis showed that the experimental molecular weights of all recombinant proteins were in excellent agreement with theoretical values (less than 1 Da), which were calculated by assuming that all cysteines form disulfide bonds. Thus, our results indicated the formation of four disulfide bonds in Ech and its mutants.

### 5.2. Mass Spectrometric Measurements

The molecular weights of Ech and its mutants were confirmed using an LTQ Orbitrap XL mass spectrometer equipped with an electrospray ionization source (Thermo Fisher Scientific, Waltham, MA, USA) as previously described [18].

### 5.3. Cell Adhesion Assay

The cell adhesion assay was performed using previously described protocols [14]. In brief, Chinese hamster ovary (CHO) cells expressing integrins αvβ3 (CHO-αvβ3) and αIIbβ3 (CHO-αIIbβ3) were kindly provided by Dr Y. Takada (Scripps Research Institute) and maintained in Dulbecco’s Modified Eagle’s medium (DMEM). Human erythroleukemia K562 cells and the human colon adenocarcinoma cell line HT-29 were purchased from ATCC. K562 cells were cultured in RPMI-1640 medium containing 5% fetal calf serum, and HT29 cells were maintained in DMEM. Ninety-six-well microtiter plates (Costar, Corning, NY, USA) were coated with 100 μL of PBS buffer containing 200 μg/mL fibrinogen (for integrins αvβ3 and αIIbβ3), 50 μg/mL fibronectin (for integrins α5β1), and 5 μg/mL vitronectin (for integrins αvβ5) at 4 °C overnight. Wells were blocked by incubating with 200 μL of heat-denatured 1% bovine serum albumin (BSA) (Calbiochem, Darmstadt, Germany) at room temperature for 40 min. The heat-denatured BSA was discarded and was washed twice with 200 μL of PBS. CHO-αvβ3, CHO-αIIbβ3, and HT29 cells were detached by trypsinization and diluted to 3 × 10^5^ cells/mL; K562 cells were diluted to 2.5 × 10^5^ cells/mL; and 105 μL of cells was used for the assay. Cells were premixed with or without Ech and its mutants for 15–30 min at 37 °C in a 5% CO_2_ atmosphere prior to dispensation into microtiter wells. The cells were then seeded to the coated plate and incubated at 37 °C (5% CO_2_) for 1 h. The non-adhered cells were removed by washing twice with 200 μL of PBS. The adhered cells were fixed with 100 μL of 3.7% paraformaldehyde and quantified using a crystal violet assay. Colorization was performed by adding 150 μL of colorizing solution containing 50% alcohol and 0.1% acetic acid. The absorbance was read at 600 nm by an automatic microplate reader. Inhibition was defined as inhibition = 100 – [OD_600_ (Rho protein-treated sample)/OD_600_ (untreated sample)] × 100%. The reported IC_50_ values are the average of at least three separate experiments.

### 5.4. Crystallization of Ech

Ech was dialyzed twice against 1 L ddH_2_O at 4 °C, lyophilized, and then dissolved in 50 mM Tris–HCl buffer (pH 8.0) at a concentration of ~10 mg/mL. Ech was initially screened using commercial kits (The JCSG Core Suites, QIAGEN, Hilden, Germany) under 384 conditions. The CombiClover Junior sitting plate (24-well, Emerald BioSystems, Bainbridge Island, WA, USA) was used for crystallization. The crystals of Ech were obtained by mixing 0.8 μL protein solution and the reservoir solution containing 2.4 M (NH_4_)_2_SO_4_ (pH 9.0) and 0.1 M Bicine at pH 9.0 by the sitting-drop vapor diffusion method. 

### 5.5. Diffraction Data Collection and Processing

All crystals were flash cooled to 100 K in a stream of cold nitrogen prior to data collection. X-ray diffraction data were collected on beamline TPS 05A at the National Synchrotron Radiation Research Center (NSRRC, Taiwan). The diffraction images were processed using the XDS program package [43]. The crystal data and data collection statistics of Ech are summarized in Table 1.

### 5.6. Structure Determination and Refinement

The initial phase of Ech was determined by molecular replacement, and the known structure of Rhodostomin (PDB ID 4RQG) in the PDB was used as our model protein. All structure determinations were carried out using alternate cycles of model building in Coot [44] and refinement in REFMAC5 [45]. The stereochemistry and structures of the final models were analyzed by MolProbity [34]. Ech belonged to the P3_2_21 space group with parameters a = 33.79, b = 33.79, and c = 120.90 Å. This structure was subsequently refined at 1.80 Å resolution using REFMAC5 to an R-factor of 0.212 and R-free of 0.255. The crystal structure of Ech was deposited in PDB ID 6LSQ.

### 5.7. NOE Analysis

The NOE experiment was conducted according to previously described protocols [14]. In brief, the backbone dynamics of Ech were determined by two-dimensional proton-detected heteronuclear NMR spectroscopy. NMR experiments were performed at 27 °C on a Bruker Avance 600 MHz spectrometer. The steady-state ^1^H-^15^N NOEs were collected from ^1^H-detected ^1^H-^15^N correlation spectra recorded with sensitivity-enhanced pulse sequences. In the NOE experiment, two spectra, one with the saturated NOE and one without, were collected. The steady-state NOEs were the ratios that were calculated between the crosspeak intensities with and without saturation of the proton resonances (i.e., NOE = I_sat_/I_nonsat_). The reported NOE value was the average value of three experiments.

### 5.8. Molecular Docking

The HADDOCK webserver was used for the docking of Ech into integrin αvβ3 [46]. The starting structure for docking was the X-ray structure of Echistatin. The interaction restraints were derived from the X-ray structure of integrin αvβ3 in complex with the 10th type III RGD domain of wild-type fibronectin (PDB ID 4MMX) [47] by using CCP4i software [48]. The defined distance threshold was 4 Å, and the interaction restraints between the RGD motif and integrin were used for the calculation. The input restraints between the R49, G50, and D51 residues of Ech and integrin αvβ3 were 27, 11, and 40, respectively. They were the contacts between the R49 residue and residues D150, Y178, Q180, and D218 of integrin αv; between residues G50 and Y178 of αv and residues R216 and A218 of β3; and between residue D51 and residues S121, Y122, S123, R214, N215, R216, D217, and A218 and Mn^2+^ of the MIDAS of β3. An additional 0.5 Å distance was defined as the minimum and maximum limits of the interaction restraints. These restraints were used to perform the standard HADDOCK protocol for protein docking with minor modifications. This protocol combines three stages of molecular dynamics calculations, including heating and cooling, with a progressive increase in flexibility at the binding interface. In the first stage, 1000 conformations were calculated using a rigid-body docking protocol. The 200 structures with the lowest intermolecular energies were refined using semiflexible simulated annealing in the second stage. Semiflexible and fully flexible regions were automatically defined by HADDOCK. The 200 structures with the lowest intermolecular energy values were refined using explicit water molecules in the last stage. The clusters of structures were identified by fitting them with an average root mean square deviation (RMSD) value of 0.5–1.5 Å for the backbone atoms of all the amino acids. The selected structure cluster for the structural analysis was based on the lowest Z-score without any restraint violations. Hydrogen bonds and salt bridges interactions were analyzed with PISA software [49].

### 5.9. VEGF-Induced HUVEC Proliferation Assay

Human umbilical vein endothelial cells (HUVECs) were purchased from Cambrex-Lonza (East Rutherford, NJ, USA). They were maintained and expanded in endothelial cell growth medium-2 (EGM-2) and gradually replaced by 20% CCS/M199. For the proliferation assay, cells were seeded in 0.2% gelatin-coated 96-well plates at 1.5 × 10^4^ cells/well, and then the cells were cultured in M199 serum-free medium at 37 °C overnight. The following day, 20 ng/mL recombinant human VEGF (R&D Systems, Minneapolis, MN, USA) was added, and the cells were treated with different concentrations of Ech for 48 h. After removing the medium, the cells of the plate were fixed with 120 μL of 3.7% paraformaldehyde and stained with 50 μL of 0.05% crystal violet solution for 20 min. Each well was then washed four times with 200 μL of distilled water and dried. Colorization was performed by adding 150 μL of colorizing solution containing 50% alcohol and 0.1% acetic acid. The absorbance was read at 600 nm by an automatic microplate reader. The reported values are the average of at least three separate experiments.

### 5.10. Cell Migration Assay

The transwell migration assay was conducted according to previously described protocols [13]. In brief, human A375 melanoma cells, human pancreatic Panc-1 cells, and U373 MG human glioblastoma cells were used to determine the inhibitory activities of recombinant Ech and its mutant. Then, 1 × 10^5^ cells were added to the transwell inserts and incubated in 24-well plates. Cells in the inserts were cultured in serum free DMEM medium, whereas the lower wells were filled with DMEM medium containing 10% FBS as an attractant. After 6 h of incubation, the non-migrated cells on the upper side of the inserts were removed using a cotton swab. The insert wells were fixed with 3.7% paraformaldehyde and stained with 0.05% crystal violet. Cells that had transmigrated to the lower surface were imaged at 100× magnification, for five random fields by using OLYMPUS-IX71 microscopy. Migrated cells were counted using NIH ImageJ software.

## Figures and Tables

**Figure 1 toxins-12-00709-f001:**
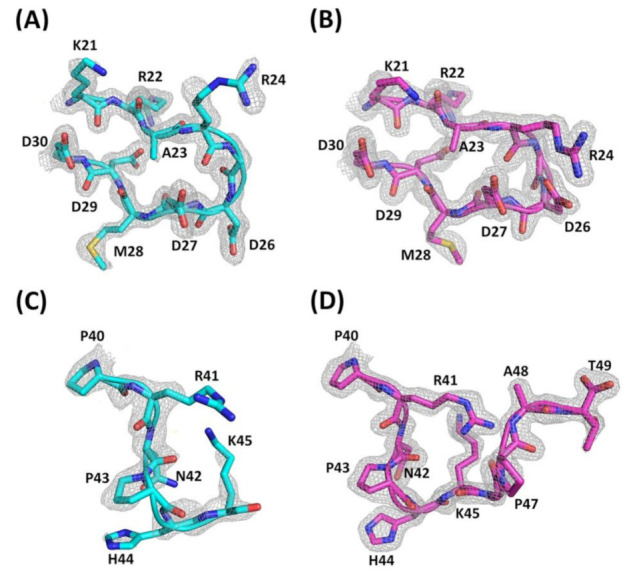
Electron density map of the RGD loop and C-termini of Ech. The 2|Fo|- |Fc| electron density map at 1.80 Å resolution is contoured at 1.0 σ around the residues from the RGD loop (K21-D30) and C-terminus (P40-T49). (**A**) Electron density map of the RGD loop with residues K21 to D30 (chain A); (**B**) Electron density map of the RGD loop with residues K21 to D30 (chain B); (**C**) Electron density map of the C-terminus with residues P40 to K45 (chain A); (**D**) Electron density map of the C-terminus with residues P40 to T49 (chain B).

**Figure 2 toxins-12-00709-f002:**
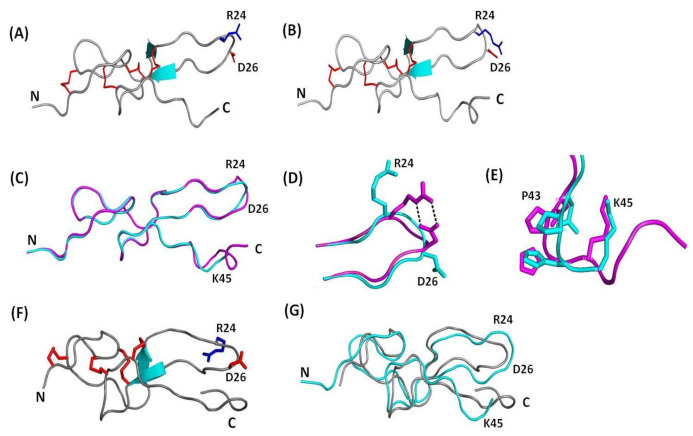
Comparison of X-ray and NMR structures of Ech. The β-sheet secondary structure is shown in cyan. The side chains of the R and D residues in the RGD loop are shown in blue and red, and disulfide bond linkages are shown in red. (**A**) Crystal structure of Ech chain A; (**B**) Crystal structure of Ech chain B; (**C**) Superimposition of Ech chain A and B, shown in cyan and magenta, respectively; (**D**) Superimposition of Ech chain A and B RGD loop with residues K21 to D30; (**E**) Superimposition of Ech chain A and B C-terminus with residues P40 to T49; (**F**) NMR structure (2Ech) of Ech; (**G**) Superimposition of Ech chain A and NMR structure (2Ech). Chain A and the NMR structure are shown in cyan and gray, respectively.

**Figure 3 toxins-12-00709-f003:**
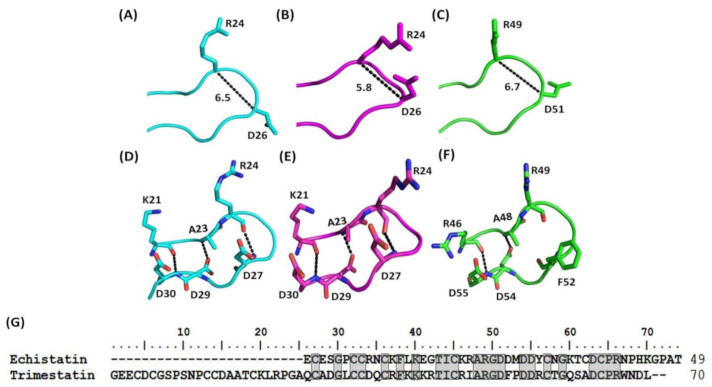
Structural comparison of the RGD loops (K21-D30) and sequence alignment of Ech and trimestatin (1J2L). (**A**) The conformation of Ech’s RGD loop (chain A). The Cα to Cα distance between residues R24 and D26 is shown; (**B**) The conformation of Ech’s RGD loop (chain B). The Cα to Cα distance between residues R24 and D26 is shown; (**C**) The conformation of trimestatin’s RGD loop. The Cα to Cα distance between residues R49 and D51 is shown; (**D**) Hydrogen bonds (residues K21-D30, A23-D29, and R24-D27) of Ech’s RGD loop (chain A); (**E**) Hydrogen bonds (residues K21-D30, A23-D29, and R24-D27) of Ech’s RGD loop (chain B); (**F**) Hydrogen bonds (residues R46-D55 and A48-D54) of trimestatin’s RGD loop; (**G**) Sequence alignment of Ech and trimestatin. The conserved residues are boxed in gray.

**Figure 4 toxins-12-00709-f004:**
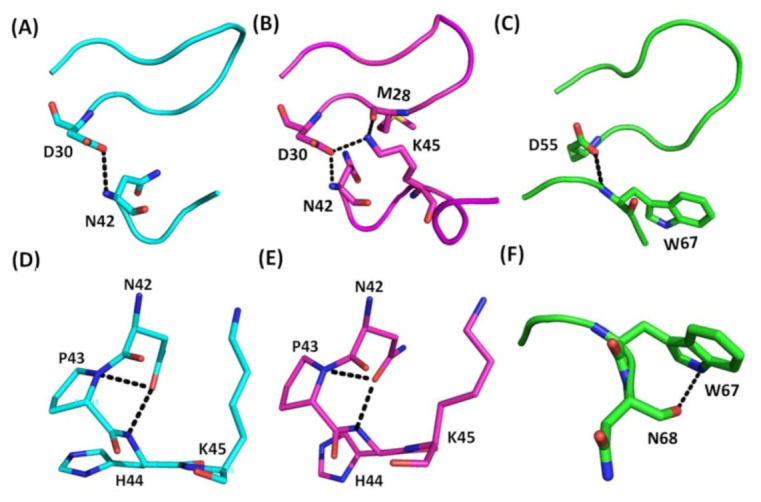
Comparison of hydrogen bonding interactions between the RGD loop and C-terminus in Ech and trimestatin, as well as those within the C-terminus. (**A**) Hydrogen bond between RGD loop and C-terminus (chain A); The hydrogen bond between the side chain of the D30 residue in the RGD loop and the amide proton of the N42 residue in the C-terminus is shown; (**B**) Hydrogen bonds between the RGD loop and C-terminus (chain B). Three hydrogen bonds between the carboxyl group of residue M28 and the sidechain Nζ proton of residue K45, between the side chain of residue D30 and the amide proton of residue N42, and between the side chain of residue D30 and the sidechain Nζ proton of residue K45; (**C**) Hydrogen bond between the RGD loop and C-terminus in trimestatin. The hydrogen bond between the side chain of the D55 residue in the RGD loop and the amide proton of the W67 residue in the C-terminus is shown; (**D**) Interaction in Ech C-terminus (chain A). The interaction between the side chain of the N42 residue and the mainchain amide proton of residues P43 and H44; (**E**) Interaction in Ech C-terminus (chain B). The interaction between the side chain of the N42 residue and the mainchain amide proton of residues P43 and H44; (**F**) Hydrogen bond in the C-terminus of trimestatin. The hydrogen bond between the side chain of residue W67 and the backbone carboxyl group of residue N68.

**Figure 5 toxins-12-00709-f005:**
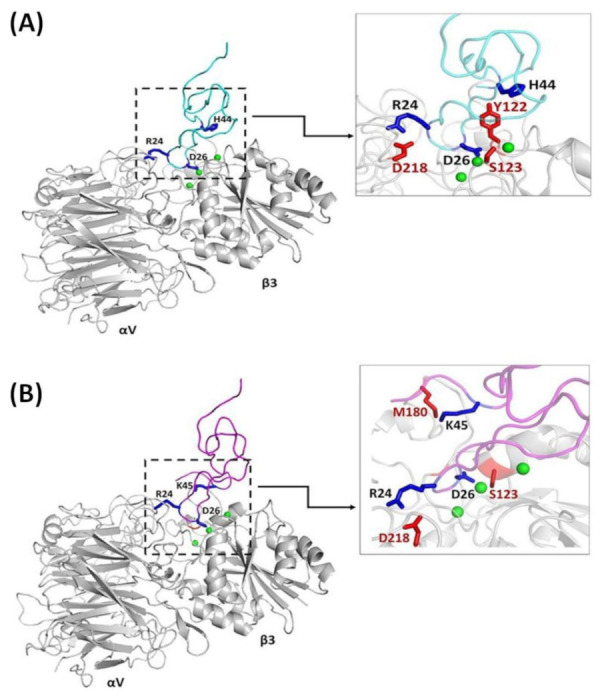
The interactions between Ech and αvβ3 integrin. (**A**) Ech chain A (cyan) docked on αvβ3 (PDB code 1L5G, gray); the main chains and side chains of the RGD loop and C-terminal interacting residues are labeled; (**B**) Ech chain B (magenta) docked on αvβ3; the main chains and side chains of the RGD loop and C-terminal interacting residues are labeled.

**Figure 6 toxins-12-00709-f006:**
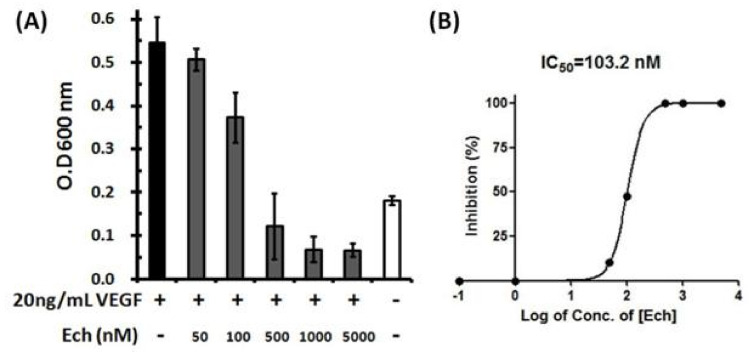
Inhibition of VEGF-induced HUVEC proliferation by Ech. (**A**) Quantitative analysis of HUVECs is shown in the histogram. The results are shown as the percentage of inhibition, and three independent experiments were performed; (**B**) A representative fitting curve from one independent experiment is shown, and the resulting IC_50_ value was 103.2 nM. HUVECs were plated in 96-well plates in the presence or absence of Ech and allowed to grow for 48 h. The cells were incubated with 20 ng/mL VEGF and then treated with 50, 100, 500, 1000, and 5000 nM Ech. The numbers of cells were quantified using crystal violet stain, and the absorbance was determined by an ELISA reader at 600 nm.

**Figure 7 toxins-12-00709-f007:**
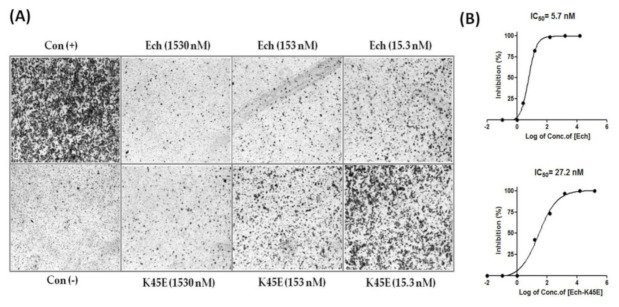
Inhibition of human glioblastoma U373 MG cell migration by Ech and its K45E mutant. (**A**) Human glioblastoma U373 MG cells were plated on a transwell filter in the presence or absence of Ech proteins and allowed to migrate for 6 h. The cells were treated with 1.53, 5.1, 15.3, 51, 153, and 1530 nM recombinant Ech proteins. The numbers of migrated cells were quantified by counting stained cells microscopically; (**B**) The results are expressed as the percentage of inhibition, and the inhibition of U373 MG cell migration by Ech and its K45E mutant was dose dependent. The mean of three independent experiments is shown.

**Figure 8 toxins-12-00709-f008:**
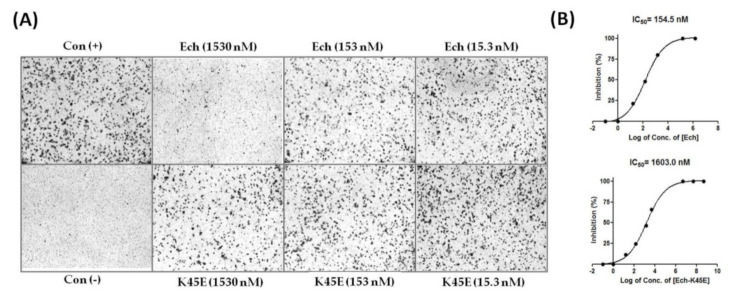
Inhibition of human pancreatic Panc-1 cell migration by Ech and its K45E mutant. (**A**) Human pancreatic Panc-1 cells were plated on a transwell filter in the presence or absence of Ech proteins and allowed to migrate for 6 h. The cells were treated with 1.53, 5.1, 15.3, 51, 153, and 1530 nM recombinant Ech proteins. The numbers of migrated cells were quantified by counting stained cells microscopically; (**B**) The results are expressed as the percentage of inhibition, and the inhibition of Panc-1 cell migration by Ech and its K45E mutant was dose dependent. The mean of three independent experiments is shown.

**Table 1 toxins-12-00709-t001:** Data collection and structure refinement.

Crystals PDB Code	Echistatin 6LSQ
**Data collection**	
Radiation source	NSRRC TPS05A
Wavelength (Å)	0.99984
Space group	*P*3_2_21
Unit cell parameters	
*a* (Å)	33.79
*b* (Å)	33.79
*c* (Å)	120.90
α	90.00
β	90.00
γ	120.00
Resolution (Å)	30.00–1.80 (1.91–1.80)
No. of reflections	7695 (1245)
Completeness (%)	92.5 (90.0)
I/σ (I)	7.88 (2.96)
CC_1/2_	99.1 (95.1)
**Refinement**	
Resolution (Å)	28.44–1.80
Reflections (work)	7541
Reflections (free)	419
R_work_ (%)	21.2
R_free_ (%)	25.5
Geometry deviations	
Bond length (Å)	0.009
Bond angles (°)	1.461
Mean B values (Å^2^)	28.851
Ramachandran plot ^1^ (%)	
Favored	95.7
Allowed	4.3

^1^ Categories were defined by MolProbity. Values in parentheses are for the highest-resolution shell.

**Table 2 toxins-12-00709-t002:** Comparison of the Cα(Ri)-Cα(Di + 2), Cβ(Ri)-Cβ(Di + 2), Cζ(Ri)-Cγ(Di + 2), and Cα(Ri)-Cα(Xi + 3) distances (Å) of echistatin and trimestatin.

Ligands	RGD Motif	Cα(Ri)-Cα(Di + 2)	Cβ(Ri)-Cβ(Di + 2)	Cζ(Ri)-Cγ(Di + 2)	Cα(Ri)-Cα(Xi + 3)
Echistatin (NMR) ^1^	ARGDDM	6.2 ± 0.75	8.2 ± 1.1	12.3 ± 0.7	6.3 ± 0.8
Echistatin (A chain)	ARGDDM	6.5	8.6	12.4	6.2
Echistatin (B chain)	ARGDDM	5.8	6.0	4.8	5.4
Trimestatin ^2^	ARGDFP	6.7	9.2	11.1	6.8

^1^ The NMR structure of Echistatin; PDB code: 2Ech. ^2^ The X-ray structure of Trimestatin; PDB code: 1J2L.

**Table 3 toxins-12-00709-t003:** Summary of the interactions between Ech chain A and αvβ3 integrin.

Ech	αvβ3	
Chain A	αv	β3
R22		N313
R24	D150, D218	
G25	D218	
D26		S121, Y122, S123, N215, R216, D217
Y31		D126
H44		Y122

**Table 4 toxins-12-00709-t004:** Summary of the interactions between Ech chain B and αvβ3 integrin.

Ech	αvβ3	
Chain B	αv	β3
R24	D150, D218	
D26		S121, Y122, S123, N215, R216, D217
M28		D126
Y31		D126
K45		M180

**Table 5 toxins-12-00709-t005:** Summary of inhibition of cell adhesion by Ech and its C-terminal truncated mutant.

Protein	Sequence				IC_50_ (nM)				
(Mutant)	RGD Loop	C-Terminus	αvβ3		αIIbβ3	α5β1		αvβ5	
Ech	^23^ARGDDM	^42^NPHKGPAT	20.7 ± 8.0		51.5 ± 3.9	132.6 ± 15.7		286.4 ± 53.6	
Ech (P43∆)	^23^ARGDDM	^42^NP	132.0 ± 12.0		362.3 ± 56.3	2469.5 ± 97.7		3358.3 ± 1040.4	

**Table 6 toxins-12-00709-t006:** Summary of inhibition of cell adhesion by Ech and its C-terminal alanine mutants.

Protein	RGD Loop	C-Terminus		IC_50_ (nM)	
(Mutant)	Sequence	Sequence	αvβ3	αIIbβ3	α5β1
Ech	^23^ARGDDM	^42^NPHKGPAT	20.7 ± 8.0	51.5 ± 3.9	132.6 ± 15.7
Ech (P43A)	^23^ARGDDM	^42^NAHKGPAT	13.8 ± 1.1	67.7 ± 35.0	58.5 ± 9.5
Ech (H44A)	^23^ARGDDM	^42^NPAKGPAT	16.4 ± 6.9	20.7 ± 4.5	30.0 ± 5.3
Ech (K45A)	^23^ARGDDM	^42^NPHAGPAT	14.7 ± 4.0	132.5 ± 53.3	84.9 ± 24.9

**Table 7 toxins-12-00709-t007:** Summary of inhibition of angiogenesis activity by Ech and other disintegrins.

Protein	RGD Motif	C-Terminal Sequence	IC_50_ (nM)
Echistatin	ARGDDM	NPHKGPAT	103.2
Saxatilin	ARGDDM	NPFHA	100.0
Salmosin	ARGDDL	NPFHA	130.0–270.0
Triflavin	ARGDFP	WNGL	100.0–400.0
Rhodostomin	PRGDMP	YH	108.0

**Table 8 toxins-12-00709-t008:** Summary of inhibition of tumor cell migration by Ech and its C-terminal mutants.

Protein	Sequence		IC_50_ (nM)		
	RGD Loop	C-Terminus	A375	U373MG	Panc-1
Ech	^23^ARGDDM	^42^NPHKGPAT	1.5	5.7	154.5
Ech (K45E)	^23^ARGDDM	^42^NPHEGPAT	13.3	27.2	1603.0

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
