# Peer review of "Structural Insight into Integrin Recognition and Anticancer Activity of Echistatin"

_toxins, 2020, doi:10.3390/toxins12110709_

Round 1

Reviewer 1 Report

Authors determined first high-resolution crystal structure of Ech, and then they had done structural analysis and performed assay.

These results are useful for both understanding Ech function and development anticancer drug.

I think this manuscript can be published after minor revision.

The minor comments were described below.

  • Line 421 … Rho (4RQG) => Rhodstomin (?) (PDB ID 4RQG)

Does Rho mean Rhodstomin? Please write proper name.

Please add “PDB ID” before 4RQG.

  • Please write PDB ID of your new structure in methods, for example line 426 in section 5.6.
  • Line 441 …(PDB code 4MMX) => (PDB ID 4MMX)

Author Response

Respond to the reviewer 1’s comments:

  1. Line 421 … Rho (4RQG) => Rhodstomin (?) (PDB ID 4RQG) Does Rho mean Rhodstomin? Please write proper name. Please add “PDB ID” before 4RQG.

As the reviewer suggested, we changed “Rho (4RQG)” to “Rhodstomin (PDB ID 4RQG)”.

  1. Please write PDB ID of your new structure in methods, for example line 426 in section 5.6.

As the reviewer suggested, we added Ech (PDB ID 6LSQ) in section 5.6 of Materials and Methods.

  1. Line 441 …(PDB code 4MMX) => (PDB ID 4MMX)

  As the reviewer suggested, we changed “(PDB code 4MMX) to “(PDB ID 4MMX)”.

Thank you very much for your assistance.

Sincerely yours,

Reviewer 2 Report

The revision authors have made is satisfactory and can be accepted for publication.

Author Response

Respond to the reviewer 2’s comments:

The revision authors have made is satisfactory and can be accepted for publication.

We thank the reviewer’s positive comment.

Thank you very much for your assistance.

Sincerely yours,

This manuscript is a resubmission of an earlier submission. The following is a list of the peer review reports and author responses from that submission.

Round 1

Reviewer 1 Report

The authors present the crystal structure of Echistatin, and some functional data on the anti-cancer properties of the molecule. The main topic of the paper is the importance of the C-terminal and RGD loop regions.

Overall, the results mostly only confirm previous observations. The crystal structure, although of high resolution, do not bring any further fundamental insights compared to the known NMR structure. In addition, the paper is mostly descriptive with listed details of the hydrogen bond network coordinating the interaction between the C-ter and RGD regions. It does not provide enough mechanistic information on the molecular interactions of Ech with integrins to help in the design of improved molecules. The authors should suggest how to achieve this based on the few mutations they carried out in this study.

In addition, the discussion mainly revolves around the network of intramolecular hydrogen bonds. However the authors do not address at all the presence of water molecules in their structure of their potential involvement. The authors need to comment on the presence of solvent and show in the figures any appropriate molecules.

Also, the authors put a lot of emphasis on the hydrogen bond interactions between K45 for example and surrounding residues. These are very flexible side chains (as illustrated by the weak electron density in chain A) and would likely be involved in protein-protein interactions with integrin when bound to it (as suggested by the K45A or K45E mutations).  Therefore the intramolecular versus intermolecular aspects need to be addressed. 

The data themselves would benefit from better presentation and link between figures (labels) and tables. 

With regards to the structure, can the author confirm that the structure factors and coordinates were deposited in the PDB, and provide a validation report?

The statistics in Table 1 should also include the CC1/2. In addition, the rms deviations for bond length/bond angles are too high (0.02Å and 2.02°) while the accepted norm should be <0.01Å and <1.5°, respectively. The structure therefore needs to be re-refine to accepted standards.

I would also recommend the authors use modern programs to re-process their data. In terms of integration and merging, HKL2000 is outdated and the authors would benefit from using either XDS or DIALS (XIA2). Also, RAMPAGE and SFCHECK are also out of use, and Molprobity statistics would be more appropriate.

Other comments:

Figure 2: please add labels of residues in stick (A-B) and indicate the N- and C-termini. Table 1: the authors should indicate what the values in brackets represent lane 44: correct 'serious' to 'series' lane 97/98: PISA offers an analysis of likelihood of biological interactions but is hardly hard proof of a monomer/dimer. Do the authors have any experimental evidence (e.g. size exclusion) lane 212: 'showed caused...' please clarify

Reviewer 2 Report

Authors determined first high resolution crystal structure of short disintegrin, and performed a cell adhesion assay and VEGF-induced HUVEC proliferation assay.

I have some concerns and comments about this work

<Major point>

(1) I think that structural study and other assays are not linked. Authors performed mutation analysis focusing on C-terminal region. But I think that this experiment could be done without crystal structure and authors did not (could not) discuss the result of mutational study based on crystal structure. Because C-terminal structures are completely difference among mol A, mol B in crystal and previously determined NMR structure. What do authors think about these points.

(2) Authors mentioned that C-terminal structure is used for developing anti-cancer regent. As mentions question(1), C-terminal region seems to be flexible. In Figure 1 (C), electron density of some residues including K45 could not be observed clearly. In figure 2 (C) and (G), C-terminal regions are not superposed well. It seems that C-terminal region would have another (unknown) functional structure for exerting anti-cancer activity. To discuss this point is very important in this paper.

<Minor>

(3) line 74-85: (2.1) this section should be moved to material and method.

(4) line 88-90: the sentences “ and belong to …. R-free of 0.242” should be moved to material and method.

(5) line 113: Authors mentioned about crystal packing effect in Figure S1. Do C-terminal regions of mol A and B interact with adjacent molecule in crystal?

(6) The orientations (view points) of figure 2(A) and (B) are different, so it is too difficult to compare. Please correct it.

(7) Figure3 is not result of author’s experiment. You should write this point clearly in the text and add reference in Figure legend. I think, it is better that you should move this figure in supplementary.  

(8) Figure 5A. and Line258: According to Figure1C, electron density of K45 could not be observed. So you cannot indicate that nitrogen atom of K45 make hydrogen bond with main chain oxygen of N42.

(9)Line 384-388: How did authors determine the initial phase? Molecular replacement?

Reviewer 3 Report

This is an interesting study done by the authors of the manuscript “Structural Insight into Integrin Recognition and Anticancer Activity of Echistatin”. Here authors have done extensive structural characterizations of wild-type and mutant Echistatin protein. They also did in vitro characterizations of wild-type and mutant Echistatin and tried to understand how mutations affects the binding of Echistatin with different integrins. Authors also did in vivo experiments in cell and found out that Echistatin inhibited VEGF-induced HUVEC cell proliferation with the IC50 value of 103.2 nM and also inhibited migration of A375, U373MG, and Panc-1 tumor cells with the IC50 values of 1.5, 5.7, and 154.5 nM, respectively. This is really interesting investigation. This study has been carried out with great care and caution, however, some minor issues demand proper attention by the authors are follows.

Materials and Methods; Cell adhesion assay: Lines-349, 352, 354, 365, 360, 362, 364; volume of the solution OR PBS buffer is written in liters for example 100 L, 200 L. Please verify this and correct if needed. I would assume this volume should be either in mL or uL. Materials and Methods; section 5.6 structure determination and refinement: Please describe the structure determination more for example which program was used, what was the phasing method, if MR used then which PDB model was used for molecular replacement. Manuscript has been written well with clear and unambiguous; and professional English. The impact of the study is high enough and suitable for the publication.